# TS2Code: Enhancing Time Series Understanding via Learning to Code

## Abstract

Despite improvements in multimodal reasoning and code generation, language models still fail to perform well on time series forecasting and reasoning. To address this, we propose **TS2Code**, a novel multi-modal training objective for learning multi-modal representation spaces for time series data. **TS2Code** works by training vision-language models to convert time series to code, which reconstructs the input time series when run. This reconstruction serves as a verifiable reward, which lets us use reinforcement learning (RL) to train models to write better code. In extensive experiments, we find that training models to convert time series into code improves their zero-shot performance on time series forecasting, anomaly detection, and reasoning, with the gains increasing with model size. In addition, by controlling code structure through RL, we further find that rewarding code styles, such as minimal digit usage, also helps improve performance.[1]

## 1 Introduction

Time series data are prevalent in various fields including finance (Carvalho et al., 1979; Peia and Roszbach, 2015), healthcare (Morid et al., 2023), human activity (Bulbul et al., 2018; Vaizman et al., 2018) and energy (Wang et al., 2024b). With the advancement of language models (LMs), leveraging LMs in conjunction with relevant context has achieved promising results on time series analyses, such as pattern understanding (Xie et al., 2024; Cai et al., 2024), anomaly detection (Liu et al., 2024d; Zhou and Yu, 2024), and time series classification (Zhang et al., 2025). LMs have been showing to possess certain zero-shot time series forecasting capabilities (Gruver et al., 2023b), and context can effectively aid time series forecasting in LMs (Wang et al., 2024c; Xu et al., 2024b; Zhang et al., 2024). However, LLMs still struggle with text-form time series reasoning (Merrill et al., 2024).

To further enhance LMs' predictive ability, several studies have constructed a representation space for time series forecasting by fine-tuning LMs on embedded time series (Liu et al., 2025b; Cao et al., 2024; Niu et al., 2025). For example, Zhou et al. (2023) fine-tune GPT-2 on encoded time series and Jin et al. (2024) employ multi-head attention to align time series embeddings with LLaMA's word embeddings. However, this approach of converting LMs into an encoder–decoder structure has limited effectiveness (Tan et al., 2024). Although Luo et al. (2025) find that fine-tuning LMs with time series in textual form within a specific domain enables forecasting, LMs face inherent difficulty in generalizing over numerical text sequences (Yang et al., 2024b). Some studies have shown LMs performing better in interpreting time series images (Liu et al., 2024a; 2025a; Kong et al., 2025a) and in downstream tasks like anomaly detection (Zhou and Yu, 2024) or reasoning (Kong et al., 2025b).

To improve LMs' ability to represent time series and better leverage their advantages in interpreting time series images, we propose **TS2Code**, which constructs a unified representation space by understanding time series through generating natural language descriptions and executable code to reconstruct the series from images, as shown in Figure 1. In contrast to building the representation space using plain text (Wang et al., 2024b; Luo et al., 2025) or time series embeddings (Kong et al., 2025a; Zhang et al., 2025; Xie et al., 2024; Wang et al., 2024a), using executable code enables a reward signal for further optimization through reinforcement learning. The code representation is also human understandable and enables interpretable time series forecasting. We find that reinforcement learning based on time series reconstruction enhances performance on downstream tasks, including

---

[1]Our implementation will be posted publicly, and is available to reviewers anonymously at `https://anonymous.4open.science/r/TS2Code-830E`.

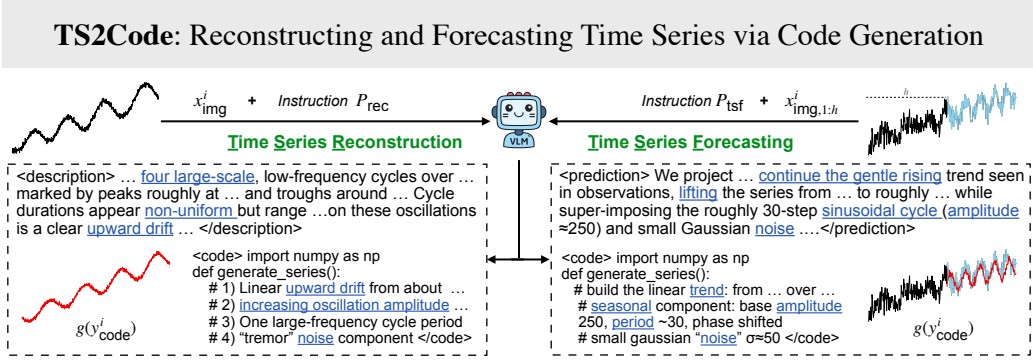

Figure 1: Illustration of time series reconstruction and forecasting via code generation. For reconstruction, which we use in reinforcement learning training, the input is a time series image, and the outputs are a description and reconstruction code that returns the time series. For time series forecasting, the input is the first $h$ steps of the series, and the output includes code that generates the predicted values. (Details can be found in Figures 13 and 14 in Appendix B.)

time series forecasting, anomaly detection, and time series reasoning. In addition, our experiments on models of different sizes (3B and 7B parameters) and modalities (vision and text) demonstrate the scalability of **TS2Code** approach, and that using images as inputs offers clear advantages over text.

**Contributions.** Our main contributions are:

- We develop a novel approach to constructing a representation space in vision-language models through natural language description and code-based time series reconstruction. (Section 4)
- We identify a reward signal with structured code generation that enables reinforcement learning to improve the representation space's understanding of time series through code-based reconstruction, rather than focusing solely on reconstruction accuracy. (Section 5.3)
- We find that code can serve as an interpretable medium for time series forecasting, and through a series of experiments, we find that the representation space optimized via code-based reconstruction improves performance on anomaly detection, time series forecasting, and reasoning. (Section 5.4)
- By exploring RL rewards, we find that digit usage in code is the factor influencing time series understanding, and that filtering poor group generations stabilizes training. (Section 5.5)

## 2 RELATED WORK

To construct effective time series representation spaces within language models (LMs), a variety of methods have been proposed, including aligning LLMs with time series embeddings or fine-tuning LLMs with contextual time series data. In addition, to enhance time series interpretation ability, many studies reveal that VLMs have a clear advantage in understanding time series through images.

**Aligning LMs with Time Series.** Gruver et al. (2023b) find that language models (LMs) possess certain zero-shot time series forecasting abilities. To optimize the time series representation space in LMs, various alignment approaches have been proposed to better align LLMs with time series data (Liu et al., 2025b; Cao et al., 2024; Pan et al., 2024; Xie et al., 2024; Wang et al., 2024a). In these methods, time series are typically projected into a one-dimensional embedding and then aligned with the language space of pre-trained LMs. For example, to enable LMs to perform better time series forecasting, Zhou et al. (2023) fine-tune language models such as GPT-2 by updating a subset of the transformer parameters using encoded time series. Similarly, Jin et al. (2024) employ a multi-head attention mechanism to align time series embeddings with LLaMA's word embeddings, after which the aligned time series are combined with the context and fed into the LMs. However, the effectiveness of approaches that use time series embeddings and convert LMs into an encoder–decoder paradigm remains limited (Tan et al., 2024).

**Fine-tuning LMs with Contextual Time Series.** Time series data are often accompanied by related textual information, where the described events can influence the future trajectory of the series. For

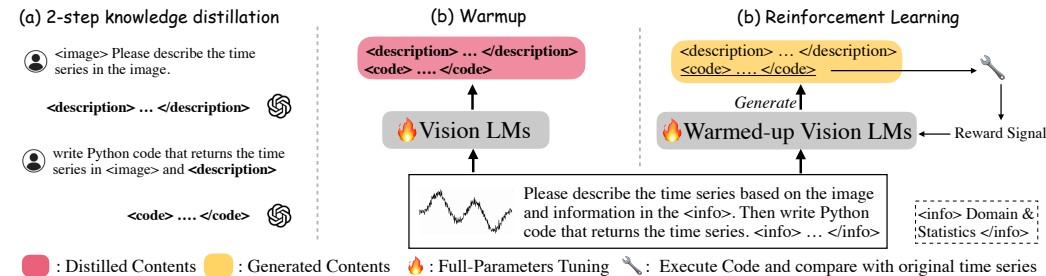

: Distilled Contents   : Generated Contents  🔥 : Full-Parameters Tuning  🔧 : Execute Code and compare with original time series

Figure 2: It is the method for constructing the representation space **TS2Code**: (a) a two-step process distills from OpenAI o4-mini both time series descriptions and executable code for reconstruction; (b) the extracted "<description>" and "" are used to supervise the training of vision-language models; and (c) reinforcement learning is applied, using reconstruction accuracy as the reward to further optimize the representation space and enhance **TS2Code** 's understanding of time series.

example, news about a cryptocurrency could affect its subsequent price (Li et al., 2024). Therefore, many studies combine event-related or domain-specific textual information with time series text to fine-tune LMs, thereby constructing a representation space that integrates time series and textual knowledge (Liu et al., 2024b;c; Kong et al., 2025a). For example, Wang et al. (2024c) combine weather forecast text with time series text to train LM, enabling it to predict solar power generation. Liu et al. (2024b) leverage "factual" information from news reports combined with time series from various domains to train LMs for assisting time series forecasting. Luo et al. (2025) leverage time series text and distilled contextual knowledge to train LMs as forecasters with reasoning processes. Similarly, Tan et al. (2025) combine text real-valued measurements (e.g., win possibility series) with events happening in matches to train LM to infer events from time series.

**Multimodal Time Series in LMs.** To improve LMs' understanding of time series, many studies leverage the multimodal capabilities of vision-language models (VLMs) to interpret time series images (Kong et al., 2025b; Liu et al., 2024a; Zhong et al., 2025; Sen et al., 2025). This is motivated by the observation that VLMs demonstrate superior performance in understanding time series patterns through images (Zhang et al., 2025; Liu et al., 2025a; Cai et al., 2024), and images continue to show advantages in downstream tasks. For example, Zhou and Yu (2024) find that LMs achieve better performance in detecting anomalies when time series are presented as images. Zhang et al. (2025) report that time series images enable more effective training of LMs for classification tasks. Similarly, Merrill et al. (2024) observe that models such as GPT4-Vision perform better on time series image reasoning than GPT-4. Therefore, in this work, we also adopt images as the input format for time series, as they are easier to interpret and more likely to generalize during training.

## 3 PROBLEM DEFINITION

Let $\mathbb{D} = \{x_{\text{series}}^i\}_{i=1}^n$ be a dataset, where each $x_{\text{series}}^i$ can be plotted as an image $x_{\text{img}}^i$, depicting a univariate time series with variable length $t_i$, and $x_{\text{series}}^i = (x_1, x_2, \ldots, x_{t_i}) \in \mathbb{R}^{t_i}$ is the corresponding values. The the **reconstruction** task aims to learn a mapping from image to original series,

$$\pi_\theta : \mathcal{X}_{\text{img}} \; \rightarrow \; \mathcal{X}_{\text{series}}$$

where the intermediate process of this mapping should demonstrate an understanding of time series in the image, such as periodicity and trends, rather than directly reading key values from the image and returning them through code for reconstruction. The code-based **time series forecasting** task differs from reconstruction in that the input time series is $x_{\text{img},1:h}^i$, which contains the first $h$ steps corresponding to values in $x_{\text{series},1:h}^i = (x_1, x_2, \ldots, x_h)$, while the goal is to learn a mapping from image to future series: $\pi_\theta : x_{\text{img},1:h}^i \rightarrow x_{\text{series},h+1:t_i}^i$.

## 4 METHODOLOGY

### 4.1 TWO-STAGE METHOD

To construct a time series representation space by generating executable code and natural language descriptions with vision-language models (VLMs), we adopt a two-stage approach. In the first stage,

given the limited capabilities of base models (e.g., *Qwen2.5-VL-3B-Instruct*), we distill knowledge from OpenAI o4-mini OpenAI (April, 2025), to warm up the base model. In the second stage, we use the comparison between the time series reconstructed from code and the original values as a verifiable reward signal for reinforcement learning to further refine the representation space.

**Warm-up Stage.** To effectively distill time series descriptions and reconstruction code, as shown in Figure 2 (a), we follow the approach of Xu et al. (2024a) and adopt a two-step method. In the first step, we prompt the model to obtain a natural language *description* of the time series using the image and basic information such as the *domain*. In the second step, we append this description to the prompt and have the LLM to generate *Python code* for reconstructing the time series. We use these description–code pairs together with instructions containing domain information and time series images, as shown in Figure 2 (b), to perform supervised fine-tuning (SFT) on the base model. This equips the base model with the preliminary ability to interpret and reconstruct time series, while also establishing connections between domain knowledge and time series in the representation space.

Therefore, the code-based time series representation process can be illustrated in Figure 1, a prompt $P_{\text{rec}}$ is used to generate the textual output: $T^i = \pi_\theta(P_{\text{rec}}, x^i_{\text{img}})$, where $T^i = \{y^i_{\text{desc}}, y^i_{\text{code}}\}$ consists of (1) a natural language description $y^i_{\text{desc}}$ and (2) *python* executable code $y^i_{\text{code}}$ for reconstruction. The numeric sequence is then obtained by executing the code: $x^i_{\text{series}} \approx g(y^i_{\text{code}})$. For details of the knowledge distillation and reconstruction instructions, refer to Figures 12 and 13 in Appendix B.

Similarly, as shown in Figure 1, the goal of time series forecasting (TSF) is to generate the future values $x^i_{\text{series},h+1:t_i}$. In our representation space, this process is carried out through code. Given a forecasting instruction $P_{\text{tsf}}$ and the input $x^i_{\text{img},1:h}$, the language model $\pi_\theta$ generates: $T^i = \pi_\theta(P_{\text{tsf}}, x^i_{\text{img},1:h})$, where $T^i = \{y^i_{\text{pred}}, y^i_{\text{code}}\}$. $y^i_{\text{pred}}$ provides a natural language prediction, while executing $y^i_{\text{code}}$ yields the predicted future sequence $\hat{x}^i_{\text{series},h+1:t_i} = g(y^i_{\text{code}})$. To equip our representation space with prelim-

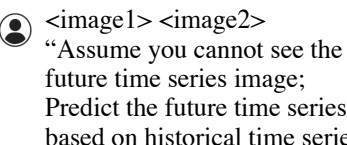

"Assume you cannot see the future time series image; Predict the future time series based on historical time series in image."

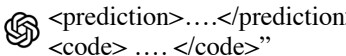

<prediction>….</prediction>
 …. "

Figure 3: TSF warm-up data distillation

inary time series forecasting ability, we additionally distilled forecasting-related knowledge, including natural language descriptions of predictions and code for generating future time series. To ensure accuracy, as illustrated in Figure 3, we provided both "historical time series" images and "future time series" images, prompting the LLM to return forecasts and code in a predictive manner, while using the future time series (ground truth) to refine prediction accuracy. For detailed instructions, see Figures 14 and 16 in Appendix B.

**Reinforcement Learning Stage.** Building on the warm-up phase, we introduce a second stage (Figure 2 (c)) based on reinforcement learning (RL) to further optimize the representation space. Here, the reconstruction accuracy is used as the reward signal to further optimize the representation space. We adopt Group Relative Policy Optimization (GRPO) (Shao et al., 2024), which samples a group of description–code pairs for a single time series and uses relative scores within the group to distinguish *high-* from *low-quality* generations, thereby guiding $\pi_\theta$ to learn from higher-quality outputs. Since after the warm-up stage our representation space has acquired a certain ability to represent time series, making it possible for $\pi_\theta$ to evolve through its own generations.

In expectation, more precise reconstruction requires LMs to develop a deeper understanding of time series features such as trends, periodicity, amplitude, and noise level, and to accurately express this understanding through code. Consequently, this also leads to better performance on evaluation tasks.

## 4.2 REWARD DESIGN FOR IMPROVING TIME SERIES UNDERSTANDING

The design of reinforcement learning rewards includes time series reconstruction **Accuracy Rewards** and **Code Quality Rewards** for controlling the style of generated code. In addition, we adopt a format reward to ensure valid outputs and apply *"Poor Group filtering"* to maintain training stability.

**Accuracy Reward.** In GRPO, for each input time series image $x_{\text{img}}$ and instruction $P_{\text{rec}}$, the model samples a group of $G$ candidate outputs $y^G_{j=1}$ (specifically, $(y^{(j)}_{\text{desc}}, y^{(j)}_{\text{code}})^G_{j=1}$), where each candidate code $y^{(j)}_{\text{code}}$ is executed to reconstruct the time series, and *accuracy reward* $r^j_1$ is computed by comparing the generated sequence with the ground-truth label $x_{\text{series}} = (x_1, x_2, \ldots, x_{t_i}) \in \mathbb{R}^{t_i}$,

$R\left(y_{\text{code}}^{(j)}, x_{\text{series}}\right), j = 1, \ldots, G$, where $R$ is defined as:

$$R(y_{\text{code}}, x_{\text{series}}) = \begin{cases} \gamma, & \text{if } g(y_{\text{code}}) \notin \mathbb{R}^{t_i} \\ \min\left(\alpha, \dfrac{\beta}{mse}\right) & mse = \text{MSE}\big(g(y_{\text{code}}), x_{\text{series}}\big). \end{cases}$$

where $g(y_{\text{code}})$ executes the generated code to obtain an array, or returns $\emptyset$ if the code fails to execute or produces an output that is not a list of numbers. When $\pi_\theta$ fails to produce a time series, a low score $\gamma$ will be assigned. The hyperparameters $\alpha$ and $\beta$ are used to control the impact of reconstruction accuracy on the reward. Mean Squared Error (MSE) denotes the metric computed on the z-score normalized time series:

$$mse = \frac{1}{t_i} \sum_{i=1}^{t_i} (x_i - x_i^{\text{rec}})^2, \, g(y_{\text{code}}) = (x_1^{\text{rec}}, x_2^{\text{rec}}, \ldots, x_{t_i}^{\text{rec}}) \in \mathbb{R}^{t_i}$$

**Code Quality Reward.** defined as $r_2^j = Q(y_{\text{code}}^j)$, are used to explore how the style of code generation affects $\pi_\theta$'s understanding of time series. **(1)** We assume that reconstruction based on higher-level time series characteristics, such as periodicity, amplitude, or trend shifts, reflects understanding-based time series generation. Therefore, $Q_1$ is used to guide $\pi_\theta$ to reconstruct using structured code rather than, as shown in Figure 15, "reading and inserting key points into the code." **(2)** We assume that using fewer digits in a single line of code reflects reconstruction through understanding time series features, therefore, $Q_2$ is used to control digit usage in a single statement. **(3)** More generally, $Q_3$ is used to control digit usage across the entire code.

**Format Reward.** $r_3^j$ is used to ensure extractable outputs, including time series descriptions wrapped in "<description>...</description>" tags and code enclosed in python code tags. We assign a score of $\lambda$ to each set of tags based on correct recognition.

Then, after collecting the rewards for each sample within the group, the group-relative advantage $A_j$ for each sample $x_{img}$ is calculated by normalizing the rewards within the group:

$$\mu = \frac{1}{G} \sum_{j=1}^{G} r_j, \quad \sigma = \sqrt{\frac{1}{G} \sum_{j=1}^{G} (r_j - \mu)^2}, \quad A_j = \frac{r_j - \mu}{\sigma}$$

**Poor Group Filtering.** In addition, we observe that for some training samples the entire group of generated codes receives very low scores. However, since GRPO adopts a group-relative advantage, normalization may convert low negative scores into positive values. However, such low-quality code is often non-executable or structurally invalid, such as an example in Figure 17, leading to training collapse (see Section 5.5 for details). To address this issue, we set group scores below a threshold to 0, ensuring that even after normalization, $A_j$ remains 0 and gradients are not updated.

## 5 EXPERIMENTS

In this section, we introduce the datasets used to build our representation space, experimental settings, and evaluation results to address the following research questions: **RQ1**: Can our **TS2Code** approach improve time series reconstruction accuracy, and does structured code generation ($Q_1$) in reinforcement learning (RL) influence code style? (Section 5.3) **RQ2**: How does RL based on time series reconstruction and $Q_1$ constraint impact performance on time series forecasting, anomaly detection, and time series reasoning? (Section 5.4) **RQ3**: How does code style, such as the use of digits in a single line ($Q_2$) or in the entire code ($Q_3$), influence the representation space? And can poor group filtering stabilize training? (Section 5.5).

### 5.1 DATASETS

To improve the representation space for time series understanding and to establish connections between time series values and domain knowledge, our training data covers a diverse range of sources, including both synthetic data and real-world data with domain-specific information, as detailed below.

**Real-world time series.** We incorporate 67 real time series datasets spanning various domains, such as weather, transportation, finance, electricity, healthcare, tourism, retail, and astronomy. These datasets are provided by the Chronos (Ansari et al., 2024) Hugging Face repository[2] and widely used for training time series foundation models Rasul et al. (2023); Das et al. (2024). The length of these time series ranges from tens to millions. For training, we extract sub-sequences with lengths from 12 to 2000, covering the requirements of most time series tasks. We manually add domain information for 67 datasets and provide it during reconstruction training with instructions.

**Time series with natural language descriptions.** Merrill et al. (2024) provide a synthetic time series dataset spanning multiple domains, where each series is accompanied by a domain description and events that affect its pattern and fluctuations. This dataset effectively enables the representation space to capture how events influence time series and to establish connections with domain knowledge. Specifically, the dataset contains 6,787 samples across approximately 1,000 domains, with sequence lengths ranging from 12 to 1,500.

**Generated synthetic time series.** Synthetic time series have benefited many studies (Xie et al., 2024; Zhou and Yu, 2024; Cai et al., 2024; Rasul et al., 2023; Das et al., 2024; Qiu et al., 2025). Similarly, to enable the representation space to capture time series patterns, we also employ synthetic time series generated through the *decomposition* method. For details of the synthetic approach, refer to Appendix C. Synthetic data also makes it possible to control learning difficulty. We categorize time series into three levels of difficulty, as illustrated in Figure 4. Simple time series (top left (a)) exhibit clear periodicity or trends. In the medium category (top right (b)), the data may include trend reversals, a mixture of periodic and trend components, or increased noise that does not obscure periodicity. In the hard category (bottom (c)), periodicity becomes irregular and noise is further amplified, potentially obscuring the underlying periodic patterns.

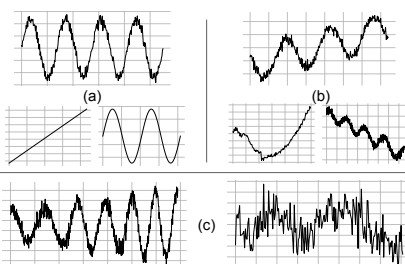

Figure 4: Synthetic time series with different levels of interpretive difficulty.

### 5.2 EXPERIMENT SETTING

**Training Data and Vision-language models.** We use $6\,000$, $2\,500$, and $4\,000$ samples from three data sources for knowledge distillation with OpenAI o4-mini (OpenAI, April, 2025). After filtering out non-executable and low-quality reconstruction (*mse* higher than 1.8), a total of $11\,500$ samples are retained for warm-up, along with $2\,000$ synthetic samples used for reinforcement learning training. Our vision-based **TS2Code** models[3] are built upon *Qwen2.5-VL-3B-Instruct* and *Qwen2.5-VL-7B-Instruct*, referred to as $\text{TSC}_{3B}$ and $\text{TSC}_{7B}$ . While the text-based models[4] are built upon *Qwen2.5-3B-Instruct* and *Qwen2.5-7B-Instruct*. The input image size is *"500x300"* with axis ticks provided. For textual time series input, following Wang et al. (2024c); Gruver et al. (2023a), we separate numeric values with commas and combine them with the instruction. For detailed training hyperparameters and further settings refer to Appendix C.

**Evaluation and Metrics.** To evaluate the improvement of **TS2Code** in time series understanding, we adopt tasks including time series forecasting (TSF), anomaly detection, and reasoning. For TSF, predicting future series from historical values requires LMs to capture features such as periodicity, trends, and scales, and to infer likely future trajectories. We sample 500 cases from synthetic data, as illustrated in Figure 4. These samples consist of **predictable patterns** without unexpected distribution shifts, but they include cases such as non-uniform periodic frequencies or noisy variations, as shown in Figure 18. For time series reasoning, etiological reasoning (Merrill et al., 2024) requires the LLM to connect time series with real-world domains, anomaly detection (Zhou and Yu, 2024) requires identifying abnormal patterns within time series, and MCQ2 (Merrill et al., 2024) requires comparing the features of two time series. Understanding time series features is a prerequisite for these tasks, so better performance implies better time series representation. For reconstruction and forecasting

---

[2]https://huggingface.co/datasets/autogluon/chronos_datasets

[3]https://huggingface.co/collections/Qwen/qwen25-vl-6795ffac22b334a837c0f9a5

[4]https://huggingface.co/collections/Qwen/qwen25-66e81a666513e518adb90d9e

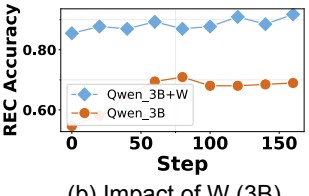
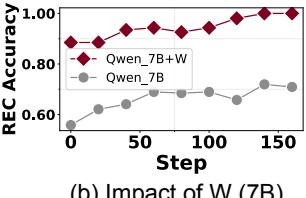

(a) $\text{RL}_{Q_1}$ Scaling (3B vs 7B)    (b) Impact of W (3B)    (b) Impact of W (7B)

Figure 5: Figure (a) compares the reconstruction performance of models of different sizes under $\text{RL}_{Q_1}$. Figures (b) and (c) illustrate the impact of warm-up (W) stage on $\text{RL}_{Q_1}$ with training steps.

accuracy, we use z-score normalized Mean Squared Error (MSE) and Mean Absolute Error (MAE) as metrics. In time series reasoning tasks, for anomaly detection, we follow Zhou and Yu (2024), converting time series into binary sequences by labeling anomalous positions as 1, and use the F1-score as the evaluation metric. For time series reasoning, we follow Merrill et al. (2024), adopting a multiple-choice setting and reporting reasoning accuracy over 500 samples.

## 5.3 IMPORVING REPRESENTATIONS THROUGH RECONSTRUCTION

To evaluate the effects of warm-up and reinforcement learning on reconstruction, we sampled 600 instances from three data sources (200 for each) introduced in Section 5.1, including 10 real-world datasets. We adopt the same training settings as described in Section 5.2 and perform reinforcement learning at least 3 times on different sets of 2000 samples.

As shown in Table 1, warm-up leads to significant improvements across the four base models, regardless of whether the time series were provided as text or images. The *confidence intervals* of accuracy can be seen in Figure 7. However, image-based models, e.g., $\text{TSC}_{3B}$ (*Qwen2.5-VL-3B-Instruct*), demonstrate a clear advantage over *text*-based model, e.g., *Qwen2.5-3B-Instruct*, under **TS2Code** training; therefore, we pri-

| Method | MAE | MSE | #inRow | #inCode | Unstrc |
|---|---|---|---|---|---|
| GPT4o (2025-01) | 0.79 | 1.37 | 21.0 | 26.3 | - |
| OpenAI o4-mini | 0.64 | 1.04 | 8.8 | 17.3 | 0.3% |
| $\text{TSC}_{3B}$ *(text)* | 0.25 | 0.48 | 95.5 | 96.2 | - |
| $\text{TSC}_{3B}$+W *(text)* | 0.84 | 1.43 | 4.5 | 17.5 | 7.0% |
| $\text{TSC}_{3B}$ | 1.02 | 1.83 | 13.4 | 17.5 | - |
| $\text{TSC}_{3B}$+W | 0.71 | 1.17 | 5.0 | 16.5 | 4.7% |
| $\text{TSC}_{3B}$+W+RL | 0.61 | 0.95 | 8.0 | 21.1 | 40.8% |
| $\text{TSC}_{3B}$+W+$\text{RL}_{Q_1}$ | 0.67 | 1.09 | 3.9 | 14.8 | 2.0% |
| $\text{TSC}_{7B}$ *(text)* | 0.17 | 0.38 | 116.2 | 117.4 | - |
| $\text{TSC}_{7B}$+W *(text)* | 0.75 | 1.24 | 5.3 | 18.2 | 7.8% |
| $\text{TSC}_{7B}$ | 0.98 | 1.79 | 3.9 | 8.9 | - |
| $\text{TSC}_{7B}$+W | 0.69 | 1.13 | 4.9 | 16.8 | 4.8% |
| $\text{TSC}_{7B}$+W+RL | 0.59 | 0.94 | 7.6 | 19.7 | 46.8% |
| $\text{TSC}_{7B}$+W+$\text{RL}_{Q_1}$ | 0.63 | 1.00 | 4.0 | 14.8 | 1.4% |

Table 1: Performance of Warm-up (**W**) and Reinforcement Learning (**RL**) in building the image-based time series representation space **TS2Code** (e.g., $\text{TSC}_{3B}$ ) and text-based space (e.g., $\text{TSC}_{3B}$ *(text)*).

marily build representation space on VLMs. Note that text-based models (e.g., $\text{TSC}_{3B}$ *(text)*) achieves lower MSE because the time series text was directly copied into the code, leading to similar values for "maximum digits in a single line" (**#inRow**) and "total digits in code" (**#inCode**).

Meanwhile, considering the case of "reading and inserting values into the code" in Figure 15, we penalize code with interpolation behavior when the statements show no explicit clues of understanding time series, such as "trend," "season," "amplitude," or functions like "sin" and "cos." Compared code generation with ($RL_{Q1}$) and without ($RL$) structured code control, we find that $Q_1$ lead to lower digits usage and proportion of unstructured code (**Unstrc**). Figure 8 (a) and (b) show word clouds of function names used in time series reconstruction on the evaluation set, providing a clearer view of the impact of structured code generation. (b) $\text{RL}_{Q_1}$ enables the representation space to employ scientific functions to understand and simulate time series rather than (a) mainly relying on *interpolation*-based methods. In addition, Figure 5 (a) demonstrates the scalability of $\text{RL}_{Q_1}$, where reconstruction accuracy (1/mse) of time series on the evaluation set improves as model parameters and data increase. Figure 5 (b) and (c) illustrate the importance of warm-up for $\text{RL}_{Q1}$.

**To conclude, the TS2Code is scalable and effective in time series reconstruction, and structured code generation ($\text{RL}_{Q1}$) provides clear evidence of its impact on controlling code style.**

| Method | Clean Pattern | | | Noised Pattern | | | Overall | | |
|---|---|---|---|---|---|---|---|---|---|
| | MAE | MSE | Err | MAE | MSE | Err | MAE | MSE | Err |
| GPT4o (*text2text*) | 0.97 | 1.61 | 6.0% | 1.03 | 1.67 | 3.6% | 1.00 | 1.64 | 4.8% |
| GPT4o (*image2text*) | 1.10 | 1.89 | 10.4% | 1.01 | 1.63 | 8.0% | 1.05 | 1.76 | 9.2% |
| GPT4o (*image2code*) | 1.05 | 1.77 | 42% | 1.12 | 1.95 | 32% | 1.09 | 1.86 | 37% |
| o4-mini (*text2text*) | **0.81** | **1.31** | **1.2%** | 1.04 | 1.71 | 8.8% | 0.93 | 1.51 | 5.0% |
| o4-mini (*image2text*) | 0.96 | 1.54 | 9.6% | 1.01 | 1.65 | 23.2% | 0.98 | 1.59 | 16.4% |
| o4-mini (*image2code*) | 0.93 | 1.39 | 3.2% | **0.87** | **1.26** | **0.4%** | **0.90** | **1.33** | **1.8%** |
| $TSC_{3B}$ | 1.05 | 1.78 | 30.0% | 1.11 | 1.98 | 29.2% | 1.08 | 1.87 | 29.6% |
| $TSC_{3B}$+W | 0.97 | 1.52 | 4.8% | 1.02 | 1.63 | 10.4% | 0.99 | 1.54 | 7.6% |
| $TSC_{3B}$+W+RL | **0.93** | 1.42 | 3.2% | 1.01 | 1.60 | 4.8% | 0.96 | 1.49 | 4.0% |
| $TSC_{3B}$+W+$RL_{Q_1}$ | **0.93** | **1.40** | **2.7%** | **0.94** | **1.41** | **1.7%** | **0.93** | **1.39** | **2.2%** |
| $TSC_{7B}$ | 1.08 | 1.86 | 42.4% | 1.08 | 1.86 | 41.2% | 1.08 | 1.86 | 41.8% |
| $TSC_{7B}$+W | 0.92 | 1.38 | **2.4%** | 1.00 | 1.59 | 3.6% | 0.95 | 1.47 | 3.0% |
| $TSC_{7B}$+W+RL | 0.91 | 1.41 | **2.4%** | 1.00 | 1.58 | **2.0%** | 0.96 | 1.48 | **2.2%** |
| $TSC_{7B}$+W+$RL_{Q_1}$ | **0.87** | **1.27** | 2.5% | **0.94** | **1.42** | **2.0%** | **0.90** | **1.34** | **2.2%** |

Table 2: The impact of Warm-up (W) and Reinforcement Learning (based on time series reconstruction) on pattern-driven time series forecasting. The results indicate that $RL_{Q_1}$ provides clear advantages in improving forecasting and reducing code execution errors. The **confidence intervals** of W and $RL_{Q_1}$ accuracy refer to Figure 9. *text2text* denotes the setting where both the input and predicted time series are in text form. *image2text* refers to using images as input and text as prediction. *image2code* uses images as input with code as the prediction medium, and **TS2Code** is evaluated on *image2code*. The above reinforcement learning results are the average inference over at least three repeated training. **Red** indicates the best performance within each slot.

## 5.4 TIME SERIES FORECASTING AND REASONING

Predicting "pattern-driven" time series requires LMs to generate future values entirely based on understanding the underlying patterns. We use the 500 cases described in Section 5.2 to evaluate the impact of reinforcement learning based on time series reconstruction on the representation space. The patterns are divided into "noisy" and "clean", and the results in Table 2 show that structured reinforcement learning ($RL_{Q1}$) performs significantly better than training without code structure constraints, with clear improvements over warm-up. The confidence intervals of accuracy can be seen in Figure 7. The gains become more pronounced as model size increases. In addition, we find that for OpenAI o4-mini, overall, using images as input and code as the prediction tool (*image2code*) achieves the best performance. However, in the setting where textual time series are used as input and future time series are predicted in text form (*text2text*), performance is better on clean patterns, while *image2code* shows clear advantages when handling noisy time series. One potential reason is VLMs generalize noise better in the image modality.

In evaluation of time series reasoning, we use three downstream tasks: (1) **Anomaly Detection**, as mentioned in Section 5.2, uses the benchmark from Zhou and Yu (2024) which includes various types of anomaly patterns such as "noisy-trend", "range", and "noisy-point". Each type contains both anomalous and normal cases, and we selected 50 test samples for each type. We follow the same instructions and metrics as provided in the benchmark. (2) For etiological reasoning (**ETI Reasoning**) and (3) **MCQ2** (reasoning over two time series), we used the datasets and settings from Merrill et al. (2024). As shown in Table 3, under the zero-shot inference setting, warm-up provides clear benefits for downstream tasks. $RL_{Q1}$ shows overall advantages compared to RL without structural control and further improves reasoning beyond the warm-up stage.

**In short, $RL_{Q_1}$ based on time series reconstruction with structured code generation ($Q_1$) provides clear improvements and advantages for tasks that rely on time series understanding.**

## 5.5 EXPLORATION OF REINFORCEMENT LEARNING

To explore the impact of code style on the representation space, as described in Section 4.2, we introduce code generation constraints $Q_2$ (maximum digit usage in a single line, **#inRow**) and $Q_3$ (digit usage across code, **#inCode**). $Q_2$ is defined as $-\alpha \cdot \max(\#inRow - 3, 0)$, where with $\alpha$

| Method | Anomaly Detection | | | ETI Reasoning | MCQ2 |
| | Precision | Recall | F1_score | Accuracy | Accuracy |
|---|---|---|---|---|---|
| GPT4o | 0.390 | 0.531 | 42.0 | 58.0% | 58.0% |
| Random | 0.033 | 0.333 | 5.7 | 25.0% | 25.0% |
| $\text{TSC}_{3B}$ | 0.148 | 0.200 | 14.7 | 25.0% | 37.2% |
| $\text{TSC}_{3B}$+W | 0.149 | 0.279 | 15.8 | 37.2% | 49.4% |
| $\text{TSC}_{3B}$+W+RL | 0.120 | 0.301 | 14.4 | 37.0% | 50.0% |
| $\text{TSC}_{3B}$+W+$\text{RL}_{Q_1}$ | 0.260 | 0.400 | **26.9** | **39.9%** | **50.3%** |
| $\text{TSC}_{7B}$ | 0.184 | 0.237 | **19.4** | 35.6% | 41.6% |
| $\text{TSC}_{7B}$+W | 0.145 | 0.423 | 17.2 | 40.8% | 54.8% |
| $\text{TSC}_{7B}$+W+RL | 0.141 | 0.522 | 17.3 | 42.0% | 54.0% |
| $\text{TSC}_{7B}$+W+$\text{RL}_{Q_1}$ | 0.140 | 0.470 | 17.0 | **45.1%** | **55.0%** |

Table 3: The impact of Warm-up (W) and Reinforcement Learning (RL) based on time series reconstruction on time series reasoning tasks. $\text{RL}_{Q_1}$ enhances reasoning ability, or at least does not introduce negative effects (e.g., in MCQ2). However, $\text{TSC}_{7B}$ produces 171 invalid answers on "Anomaly Detection," while +W reduces this to 0, even though accuracy does not improve. The above reinforcement learning results are the average inference over at least three repeated training. The input image size for anomaly detection is $1000 \times 300$, while the MCQ2 task uses two input images.

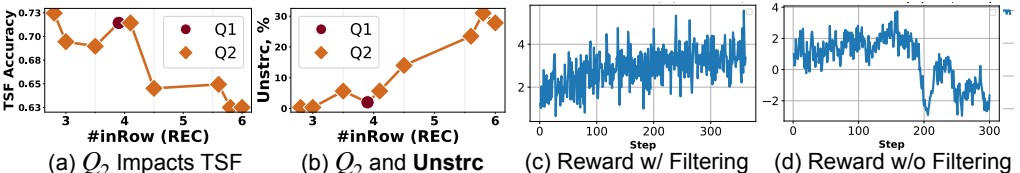

(a) $Q_2$ Impacts TSF    (b) $Q_2$ and **Unstrc**    (c) Reward w/ Filtering    (d) Reward w/o Filtering

Figure 6: (a) illustrates the relationship between **#inRow** (the average over the reconstruction evaluation set for each style-specific model trained with $\text{RL}_{Q_2}$) and performance on the time series forecasting (TSF) task. Similarly, (b) is about the connection between **#inRow** and unstructured code (**Unstrc**). Figures (c) and (d) compare the changes in reward over steps after filtering poor groups.

increasing in training, the model uses fewer **#inRow** (on average) in the time series reconstruction test set. As shown in Figure 6 (a), fewer digits used in a single line of code lead to better time series forecasting accuracy ($1/mse$). One potential reason is that structured code generation essentially restricts the number of digits used in a single line, thereby preventing the direct "reading and insertion of values into code" as shown in Figure 15. Moreover, in (b), we observe a reduction in the proportion of unstructured code. However, as shown in Figure 11, the changes in **#inCode** caused by $Q_3$ do not show a clear relationship with time series understanding. In addition, Figure 6 (d) is about the ablation of *Poor Group Filtering* in (c), where entire groups of low-quality generations are filtered out (excluding them from gradient updates). As a result, the trajectory of the *accuracy reward* becomes more stable with steps. An example of a low-quality generation that nevertheless receives a relatively high reward within its group can be found in Figure 17 in Appendix B.

**In short, optimizing the representation space through code-based time series reconstruction is impacted by code style, such as digit usage. Filtering out poor groups makes training stable.**

# 6 CONCLUSIONS AND FUTURE WORK

**TS2Code** is shown to be an effective method for constructing a time series representation space within vision-language models, where code-based reconstruction provides verifiable reward signals for reinforcement learning to further optimize this space. Code also can serve as a more interpretable medium for time series forecasting. In further evaluations, we find that the representation space optimized by **TS2Code** benefits time series reasoning, forecasting, and anomaly detection. In addition, we explore the impact of code style on the representation space by controlling code generation through reinforcement learning and find that digit usage affects the understanding of time series. In future work, we will further explore the potential of this representation space in time series task-specific training, such as context-aided time series forecasting.

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

**TS2Code: Enhancing Time Series Understanding via Learning to Code (Appendix)**

## A STATEMENT ON THE USE OF LLMS TOOLS

We used AI tools (LLMs) to a limited extent for polishing a part of the writing, and for generating parts of the Python plotting functions in the figure code.

## B INSTRUCTION AND CASE STUDIES

**Time Series Reconstruction.** The reconstruction instructions and case study are shown in Figure 13. Statistical information in the prompt, such as Length, is replaced according to different time series, while domain information in "<info>" is substituted based on the domain of the time series data. In addition, in the training data, we use GPT-4o (2025-01-01) to rewrite instructions to increase their diversity without altering their meaning. In the case study, the output of $\text{TSC}_{7B}+\text{W}+\text{RL}_{Q_1}$ shows that the representation space captures the features of the time series and reconstructs them through understanding these features. Note that the input images are of size 500×300 pixels, and axis and scale information are provided. In addition, the distillation prompt for warm-up data can be seen in Figure 12. This is a two-stage approach: in the first stage we obtain the description, and in the second stage, with the help of the description, we obtain the reconstruction code.

**Code-Based Time Series Forecasting.** The code-based time series forecasting instructions are shown in Figure 14, where the statistical information in the instructions is replaced according to different time series. Since synthetic data are used for the forecasting task, the domain descriptions in <info> are also synthetic. We provide only the historical part of the time series image (the **black segment**) as input. The image size is 500×300, with axes and scale information included. In the case study, we observe that the representation space outputs both natural language descriptions of the time series forecast and code-based simulations. However, inconsistencies may occur between the natural language and code descriptions (highlighted in red), which we aim to address in future work. More examples of time series forecasting performance are shown in Figure 18.

**Reading and Inserting Key Points into the Code.** The example in Figure 15 shows that the representation space reconstructs time series by reading key points. This approach has been shown not to enhance the representation space's ability to understand time series, even if it yields higher reconstruction accuracy (Section 5.3 and 5.4), and we avoid it through structured generation $\text{RL}_{Q_1}$. In exploring the impact of code style on the representation space, we find that digit usage in code is related to this type of generation (Section 5.5).

**Instruction of Time Series Forecasting Distillation.** Due to the fact that the original models, such as *Qwen2.5-VL-7B-Instruct*, do not have the ability to use code for time series forecasting. We distill another 3,000 code-based time series forecasting samples to the warm-up training set. Figure 16 illustrates knowledge distillation for time series forecasting. We provide historical time series images for OpenAI o4-mini (OpenAI, April, 2025) to generate natural language predictions within the <prediction> tag, as well as code to produce the future time series. In addition, we also provide the future time series images to allow the distillation source to reference them when generating code, thereby improving code quality (forecasting accuracy). However, the entire process is framed in a predictive manner, with the assumption that the future time series is not given.

**Case study of poor generation.** Poor group generation refers to cases where all outputs within a group are of low quality, however, an ineffective time series output receives a relatively high reward. Figure 17 shows such a (relatively) high-scoring output within a poor group. After GRPO normalization, the originally low reward becomes positive, causing the representation space to learn from low-quality code and leading to training collapse. By filtering out these groups (setting the entire group's reward to 0), training becomes more stable (Section 5.5).

## C EXPERIMENTAL DETAILS

### A LANGUAGE MODELS AND TRAINING SETTINGS

**OpenAI o4-mini (OpenAI, April, 2025).** In our knowledge distillation, we select o4-mini as the distillation source. As shown in Table 1 for time series reconstruction and Table 2 for time series

forecasting, it performs significantly better than GPT-4o. One possible reason is that reasoning-oriented models are more proficient at code generation.

**GPT4o (Achiam et al., 2023).** We use the GPT-4o (2025-01-01) API as a benchmark and reference for time series reconstruction and other tasks.

**Qwen2.5 (Vision) (Bai et al., 2025) .** For vision language models, we use the Qwen2.5-VL-3B-Instruct[5] and Qwen2.5-VL-7B-Instruct[6] to evaluate the effectiveness and scalability of **TS2Code**.

**Qwen2.5 (Text) (Yang et al., 2024a).** To evaluate the effectiveness of textual time series input, we use Qwen2.5-3B-Instruct[7] and Qwen2.5-7B-Instruct[8], two language models designed for text input.

**Hyperparameters.** In the warm-up stage, we use LLaMA-Factory Zheng et al. (2024) for supervised fine-tuning (SFT). We perform full-parameter tuning using two Nvidia H200s with a learning rate of 2e-6, a batch size of 1 with gradient accumulation of 2 and 4 epochs. For reinforcement learning, we adopt the Open-R1-based Face (2025) vision framework,VLM-R1 Shen et al. (2025), utilizing 6 Nvidia H200 and generating 8 samples per input with gradient accumulation of 2. The KL divergence coefficient $\beta$ was set to 0.04 and learning rate of 5e-6. Further details can be found in our repository[9].

**Synthetic Time Series.** We generate synthetic time series using a decompositional additive approach, with the synthesis code outlined in Algorithm 1. The series length is set between 48 and 1024 to cover most time series tasks, followed by the introduction of trend shifts, varying periodicities, and noise. Finally, the value range of the time series is set between -2000 and 5000 to cover a wide range of real-world time series scales. For time series forecasting, we adopt a similar synthesis strategy but focus primarily on **"predictable"** patterns. For example, distribution shifts are excluded from the test set, while non-uniform periodicities are included, with examples shown in Figure 14 and 18. The full implementation of our synthetic time series generation is shared in our repository.

## B ADDITIONAL EXPERIMENTAL RESULTS

**Time Series Reconstruction.** The 95% confidence interval results for time series reconstruction are shown in Figure 7. Both $\text{RL}_{Q_1}$ and reinforcement learning without structured code generation constraints provide clear improvements in reconstruction accuracy. With the "reading and inserting key points into the code" generation mode, RL even outperforms $\text{RL}_{Q_1}$ on time series reconstruction.

**Function Name Word Clouds.** Figure 8 compares the word clouds of function names (with font size indicating frequency of functions) used in time series reconstruction on the test set (600 test cases) between reinforcement learning (RL) without structured code constraints and $\text{RL}_{Q_1}$. We observe that structured code generation $Q_1$ imposes clear restrictions on function usage, shifting from Figure 8 (a), which mainly relies on the "np.interp" function for reading and interpolation (as shown in the example in Figure 15), to (b) using functions such as *sin* to simulate time series. The function name word cloud for OpenAI o4-mini and GPT4o (2025-01-01) can be seen in Figure 10.

**Time Series Forecasting.** The performance of $\text{TSC}_{3B}$ and $\text{TSC}_{7B}$ under warm-up and reinforcement learning based on time series reconstruction is shown in Figure 9. The 95% **confidence intervals demonstrate that $\text{RL}_{Q_1}$ provides clear advantages, and these gains become more pronounced as model size increases from 3B to 7B**. In contrast, reinforcement learning without structured code generation constraints, although achieving better time series reconstruction in Figure 7, does not lead to significant improvements in time series understanding.

In addition, we include statistical methods such as mean prediction (using the average of historical values as the forecast for all future values) and ARIMA, as well as foundation models such as Chronos and TimesFM, to evaluate forecasting performance on our dataset. As shown in Table 4, ARIMA (using the "*auto_arima*" function from "*pmdarima*") performs well on noise-free time

---

[5]https://huggingface.co/Qwen/Qwen2.5-VL-3B-Instruct

[6]https://huggingface.co/Qwen/Qwen2.5-VL-7B-Instruct

[7]https://huggingface.co/Qwen/Qwen2.5-3B-Instruct

[8]https://huggingface.co/Qwen/Qwen2.5-7B-Instruct

[9]Our code about detailed training settings and time series synthetic are accessible in `https://anonymous.4open.science/r/TS2Code-830E`

---

**Algorithm 1** Synthetic Time Series Generator

---

1: Initialize the random seed and sample the time-series length $L \in [48, 1024]$;
2: Enable complex trend: setting the binary flag *multi_trend* $\in \{1, 0\}$.
3: **if** *multi_trend* **then**
4:    **Shifting *trend*:** Introduce a trend shift near the midpoint of the series, i.e., at $t \approx 0.5L$.
5: **else**
6:    **Monotonic *trend*:** specify a single-regime trend that remains monotone over the entire series.
7: Enable complex seasonal period: setting the binary flag *variable_period* $\in \{1, 0\}$.
8: **if** *variable_period* **then**
9:    **Varying *seasonality*:** construct a seasonal component with non-uniform periodicity, allowing the effective frequency to change smoothly over time.
10: **else**
11:    **Uniform *seasonality*:** construct a seasonal component with a fixed period, maintaining constant frequency across the entire series.
12: **Synthesize** the raw series by decompositional addition of its constituent components (trend, seasonality, and noise). *raw_series = trend + seasonal + noise*.
13: **Rescaling:** Linearly map *raw_series* to the target range $[-2000, 5000]$.
14: **Output:** Return the rescaled series.

---

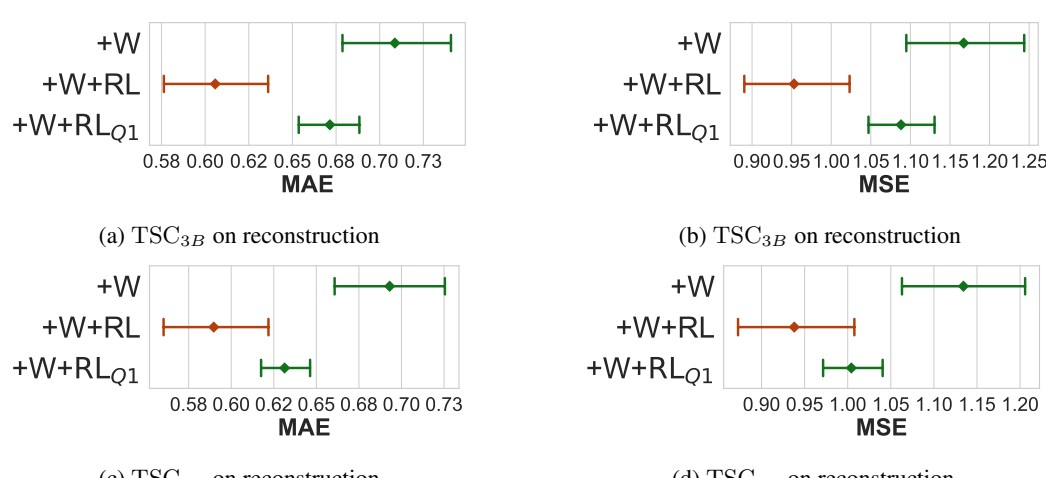

(a) $\text{TSC}_{3B}$ on reconstruction  (b) $\text{TSC}_{3B}$ on reconstruction

(c) $\text{TSC}_{7B}$ on reconstruction  (d) $\text{TSC}_{7B}$ on reconstruction

Figure 7: The overall MAE and MSE of $\text{TSC}_{3B}$ and $\text{TSC}_{7B}$ on time series reconstruction evaluation set under different training conditions (with 95% confidence intervals). The performance of $\text{TSC}_{3B}$ with warm-up (W) and $\text{RL}_{Q_1}$ is shown in (a) and (b). $\text{TSC}_{7B}$ is in (c) and (d). As the model size increases (from 3B to 7B), the benefits of reinforcement learning become more pronounced.

series, while TimesFM shows significantly better performance on noisy time series. Prediction cases of $\text{TSC}_{7B}\text{+W+RL}_{Q_1}$ and other foundation models on noised and clean patterns can be seen in Figure 18.

**Code generation constraint.** $Q_3$ is defined as $-\alpha \cdot \max(\#\text{inCode} - 8, 0)$ and is used in reinforcement learning to penalize digit usage in time series reconstruction code. We find that code generation style (the maximum number of digits used in a single line of code, **#inRow**) is related to the representation space in time series forecasting (Section 5.5). However, Figure 11(a) shows the representation space trained under different levels of $Q_3$ constraints. Using fewer digits in code (**#inCode**) on the time series reconstruction (REC) test set does not affect time series forecasting performance, even though (b) shows that **#inCode** remains correlated with structured code on reconstruction task.

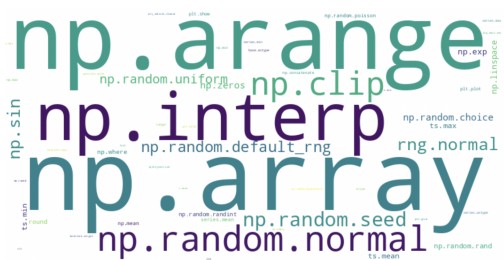

(a) Representation Space trained without $Q_1$

(b) Representation Space trained by $\mathrm{RL}_{Q_1}$

Figure 8: The function name word cloud on the time series reconstruction evaluation set (with font size indicating frequency of functions). (a) is the representation space trained without structured code generation, $Q_1$. While, (b) is representation space trained by $\mathrm{RL}_{Q_1}$.

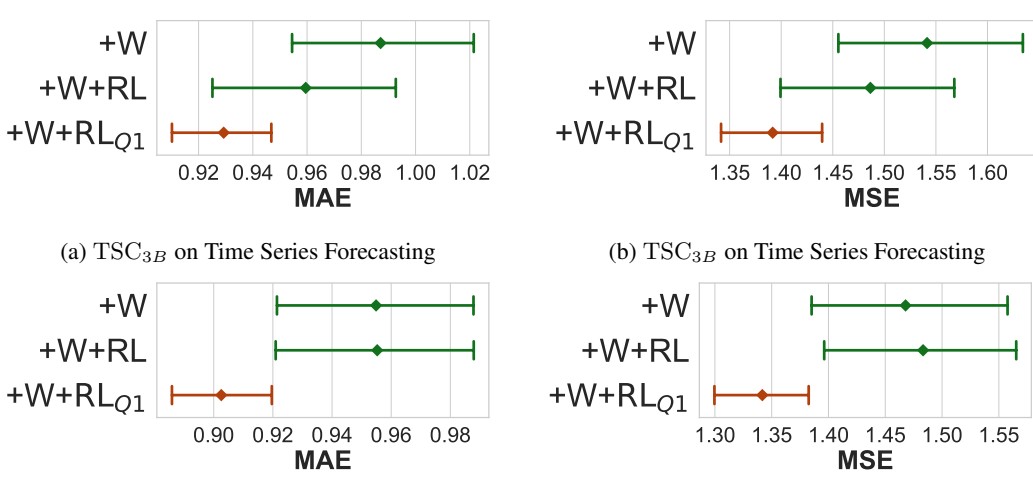

(a) $\mathrm{TSC}_{3B}$ on Time Series Forecasting

(b) $\mathrm{TSC}_{3B}$ on Time Series Forecasting

(c) $\mathrm{TSC}_{7B}$ on Time Series Forecasting

(d) $\mathrm{TSC}_{7B}$ on Time Series Forecasting

Figure 9: The overall MAE and MSE of $\mathrm{TSC}_{3B}$ and $\mathrm{TSC}_{7B}$ on time series forecasting (TSF) evaluation set under different training conditions (with 95% confidence intervals). $\mathrm{RL}_{Q_1}$ shows clear advantages, and as the model size increases, the benefits of $\mathrm{RL}_{Q_1}$ become more pronounced.

| Method | Clean Pattern | | Noised Pattern | | Overall | |
|---|---|---|---|---|---|---|
| | MAE | MSE | MAE | MSE | MAE | MSE |
| Mean Prediction | 1.04 | 1.66 | 1.06 | 1.68 | 1.05 | 1.67 |
| Chronos (Base) (Ansari et al., 2024) | 0.77 | 1.25 | 1.01 | 1.67 | 0.89 | 1.46 |
| Chronos (Large) (Ansari et al., 2024) | 0.76 | 1.19 | 0.99 | 1.64 | 0.87 | 1.42 |
| ARIMIA | 0.70 | 1.12 | 0.91 | 1.45 | 0.81 | 1.28 |
| TimesFM (Das et al., 2024) | 0.86 | 1.36 | 0.48 | 0.74 | 0.67 | 1.05 |

Table 4: Performance of foundation models and simple baselines on our time series forecasting dataset. Here, mean prediction uses the average of historical values as the forecast for all future values, while ARIMA is implemented using the "*auto_arima*" function from "*pmdarima*". Red font indicates the best performance in the column.

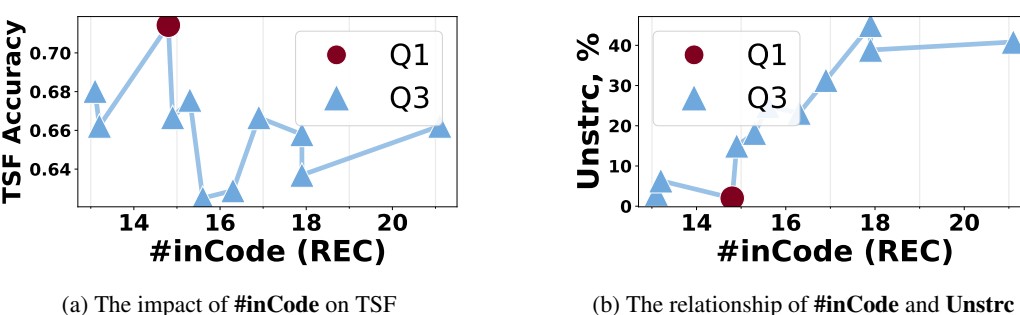

(a) OpenAI o4-mini

(b) GPT4o (2025-01-01)

Figure 10: The function name word cloud on the time series reconstruction evaluation set (with font size indicating frequency of functions). (a) is from OpenAI o4-mini (OpenAI, April, 2025). While, (b) is from GPT4o (2025-01-01).

(a) The impact of **#inCode** on TSF

(b) The relationship of **#inCode** and **Unstrc**

Figure 11: Under $Q_3$ control, the code style in the representation space ($\text{TSC}_{3B}$) on the time series reconstruction test set (measured by the total number of digits in code, **#inCode**) is constrained. (a) shows the relationship between **#inCode** and time series forecasting accuracy ($1/mse$), while (b) shows the relationship between **#inCode** and the proportion of unstructured code during reconstruction (**Unstrc**).

---

**Two-stage knowledge distillation prompt for reconstruction**

**Stage-1**
Please incorporate information in "<info>" to briefly describe the time series shown in the image.
<info>{intro}</info>

**Stage-2**
Based on the provided time series image, along with the information in the "<info>" and "<description>" tags, please write a Python function "generate_series()" that returns the time series as a NumPy array.
<info>{intro}</info>
<description>{description}</description>
Put your code in this format:
```python
def generate_series():
# your generation code here
```

---

Figure 12: Two-stage knowledge distillation prompt for time series reconstruction from OpenAI o4-mini. In Stage 1, we provide the time series image and metadata (domain and summary statistics). In Stage 2, we augment the prompt with the description obtained in the previous round and additional information, prompting the LLM to return reconstruction code in a Python function.

## Instruction of Time Series Reconstruction

For the provided time series image and the information in "<info>", please first describe it. Then, based on your understanding, write a Python function "generate_series()" that returns the time series as a NumPy array.
<info> This is a synthetic time series. Please reconstruct it based on the features you observe, such as periodicity, peaks and valleys, trends, and noise. Length: 825, max: 1390.2, min: -313.0, mean: 516.2 </info>
Put your description in this format:
<description> ... </description>
Put your code in this format:
```python
def generate_series():
# your code here
```

## Case Study: Output of Time Series Reconstruction

<description> The series exhibits about **four large-scale, low-frequency cycles** over its 825-step span, marked by **peaks** roughly at steps 150, 310, 490, and 685 and **troughs** around steps 70, 240, 415, and 605. **Cycle durations appear non-uniform** but range on the order of 110–160 steps between peaks, yielding $\sim$ 2 cycles - 1/2 cycles per 200-step interval. Superimposed on these oscillations is **a clear upward drift**: early troughs sit below –280 and peaks near 165, whereas in the final cycles troughs fall around 620–690 and peaks creep above 1,260. Both peaks and valleys trend **steadily upward** across the series, lifting both the baseline and the amplitude of each cycle over time. </description>

```python
import numpy as np
def generate_series():
""" Generate a synthetic time series of length 825 with 1) ~ 4 low-frequency oscillations 2) an upward drift 3) growing amplitude Approximate $min \approx -300, max \approx +1,350$, mean around 500 """
N = 825
t = np.arange(N)
# 1) Linear upward drift from about -200 up to ~ 1100
drift = np.linspace(-200, 1100, N)
# 2) Slowly increasing oscillation amplitude (small peaks at start, larger at end)
amp_start, amp_end = 120.0, 200.0
amplitude = np.linspace(amp_start, amp_end, N)
# 3) One large-frequency cycle period
period = 200.0
# Phase shift so that our biggest peaks sit at $t \simeq 150, 310, 490, 685$
phase_shift = np.pi/2 - 2*np.pi * 150/period
oscillation = amplitude * np.sin(2*np.pi*t/period + phase_shift)
# 4) Small "tremor" noise component
rng = np.random.RandomState(42)
noise = rng.normal(scale=20.0, size=N)
series = drift + oscillation + noise
# 5) Recenter to desired mean $\simeq$ 516.2
series = series - series.mean() + 516.2
return series
```

Figure 13: The above case study illustrates how the representation space trained with **TS2Code** performs time series reconstruction along with task instructions. The outputs include a natural language description and code that generate the reconstructed time series values from the image. **Green Text** highlights clues on how the representation space understands time series patterns.

**Instruction Time Series Forecasting**

Please forecast the next 142 data points based on the 161-step historical time series in the image and the information provided in <info>. If <info> does not contain signals about future changes, please base your prediction on the historical pattern. Put your prediction in <prediction>, and write a Python function generate_series() that returns your predicted future series as a NumPy array.
<info> This is just a synthetic time series. Historical Length: 161. The last value in history: 2340.2. Future Length: 142. </info>
Put your prediction in:
<prediction>...</prediction>
Put your code in:
```python
def generate_series():
# your code here
```

**Case Study: Code-Based Time Series Forecasting**

<prediction>
We project that the next $\approx 142$ points will continue the gentle rising trend seen in the first 161 observations, lifting the series from about 2340 at the first forecast step up to roughly 2800 by the last forecast step, while super-imposing the roughly 30-step sinusoidal cycle (amplitude $\approx 250$) and small Gaussian noise ($\sigma \approx 50$). Concretely, if t is the total time index ($t = 0 \ldots 161$ historical, $t = 161 \ldots 299$ future), our forecast is

$y_t \approx 2340.2 + (2800 - 2340.2)/161 \cdot t + 250 \cdot sin[2\pi \cdot (t - 30)/30] + \epsilon_t$

with $\epsilon_t \sim N(0, 50^2)$.
</prediction>
```python
import numpy as np
def generate_series():
"""Generate the next 142 points of the synthetic time series by continuing the trend and repeating the same  30-step seasonal cycle from the history. """
N = 142
# time index for the future points
t = np.arange(N)
# end-values at the first and last forecast steps
y0 = 2340.2
y_end = 2800.0
# build the linear trend: from 2340.2 to 2800 over N-1 steps
trend_slope = (y_end - y0) / (N - 1)
trend = y0 + trend_slope * t
# seasonal component: base amplitude 250, period  30, phase shifted
seasonal = 250.0 * np.sin(2.0 * np.pi * (t + 30.0) / 30.0)
# small gaussian "noise" σ ≈ 50
# the following produces increments of approximately 0 mean,σ ≈ 50
noise = np.random.default_rng(1234).normal(loc=0.0, scale=50.0, size=N)
series = trend + seasonal + noise
return series
```

Figure 14: The above case study illustrates how the representation space trained with **TS2Code** performs time series forecasting along with task instructions (black indicates the input historical time series, and blue denotes the portion to be predicted). The outputs include a natural language description of the forecasting process enclosed in "<prediction>" and code that generates the predicted time series values. **Green Text** highlights some of the reasoning steps taken by the language model, indicating how it uses code to perform time series forecasting. While **Red Text** marks inconsistencies between the natural language description and the code, which represent limitations of **TS2Code** that we aim to address in future work.

**Case Study: Reconstructing Time Series by Reading and Inserting Key Points**

<description>
The 976–step series is dominated by about four large-scale oscillations whose lengths slowly drift upward. **Key peaks occur near steps 97, 331, 592 and 815, with corresponding lows around 217, 458, 715 and 928**, .... so beyond the oscillation there is an overall rising trend.
</description>
```python
import numpy as np
def generate_series():
    """
    Generate a 976-step time series exhibiting  4 broad oscillations with slowly widening period,
    an upward drift, a rising baseline, and small-scale irregularity plus noise. Returns ——- series
    : np.ndarray, shape (976,) The synthetic time series.
    """
    N = 976
    t = np.arange(N)
    # Key (index, value) pairs: the major peaks and valleys
    key_idx = np.array([0, 97, 217, 331, 458, 592, 715, 815, 928, 975])
    key_val = np.array([-850, -265, -536, 307, -525, 516, -28, 795, -177, 450])
    # 1) Baseline by linear interpolation through these knots
    baseline = np.interp(t, key_idx, key_val)
    # 2) Slowly increasing period: from  228 steps up to  297
    period = 228 + (t / (N - 1)) * 70 # 70 steps worth of period drift
    freq = 2 * np.pi / period
    # 3) Build the core sine-wave oscillation
    # we choose a cosine so we get peaks at the prescribed t_peak
    phase = 2 * np.pi * (t - 97) / 228.0
    oscillation = np.cos(phase)
    # 4) Add some extra small-scale "jitter" / noise
    noise = np.random.normal(loc=0.0, scale=10.0, size=N)
    # Combine
    series = baseline + oscillation * 30 + noise
    return series
```

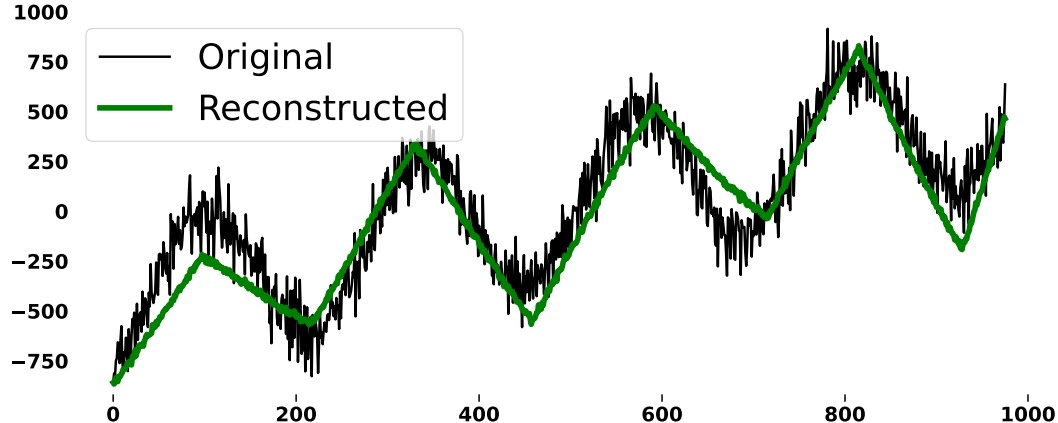

Figure 15: Case A illustrates an example of "Reconstructing Time Series by Reading Key Points." Instead of understanding patterns in the time series—such as periodicity or fluctuations—to generate the series, the language model reads key points from the time series image and embeds them directly into the code for reconstruction. Although this approach yields relatively high reconstruction accuracy, the evaluation in Section 5 shows that it does not improve time series understanding. **The Green Text** indicates the reasoning process of a language model reconstructing by reading key points into the code, while the **Green Line** in the figure shows the reconstructed time series.

---

**Instruction of Time Series Forecasting Distillation**

You are required to complete the following task from the perspective of forecasting a future time series.
Please forecast the future {fut_len} data points (In the future time series image) based on the {his_len}-step historical time series image (In the history time series image) and the information provided in <info>. Finally, write a Python function generate_series() that returns the predicted future series as a NumPy array.

**Note that when describing your prediction, assume you have not seen the future time series image.** However, when making the actual prediction and writing the code, you may refer to the true future values to improve your accuracy.
{info}
Put your prediction in:
<prediction> ... </prediction>
Put your code in this format:
```python
def generate_series():
# your code here
```

Figure 16: It illustrates the instruction for distilling time series forecasting warm-up data. Two images are provided: the first shows the historical time series, and the second corresponds to the portion to be forecast. The LLM is prompted to return prediction text descriptions and code in a predictive manner, while the ground-truth time series image is used to improve the accuracy of the returned predictions.

---

**A case study of Poor Group generation**

```python
<description>...</description>
import numpy as np
def generate_series():
""" Generate a synthetic 813-step time series that mimics the prescribed periodic oscillations,
rising trend, and noise. Returns ... """
np.random.seed(0) # Optional: for reproducibility
N = 813
x = np.arange(N)
# 1) Linear rising trend from   -20 up to   680
trend = -20 + (460.0 / (N - 1)) * x
```
**much cycles tech hints here fluently.**
```
cyclist fluxy noncycle fixes fluv.
nonfix explicit reps not exact:
cycle word explicit:
just think align exact better fix fixcycles noncert exact digicalNon cycle?
system
template cycle noncastNonRollulativeNonFeb:
global better flu:
N fact fix cycles:
but nearer bettercycle hyporexlikely:
huge non
...
```

Figure 17: This excerpt is a relatively high-scoring output case from poor group generation. Starting from the Red Text, the LLM begins to produce random content without returning a valid time series.

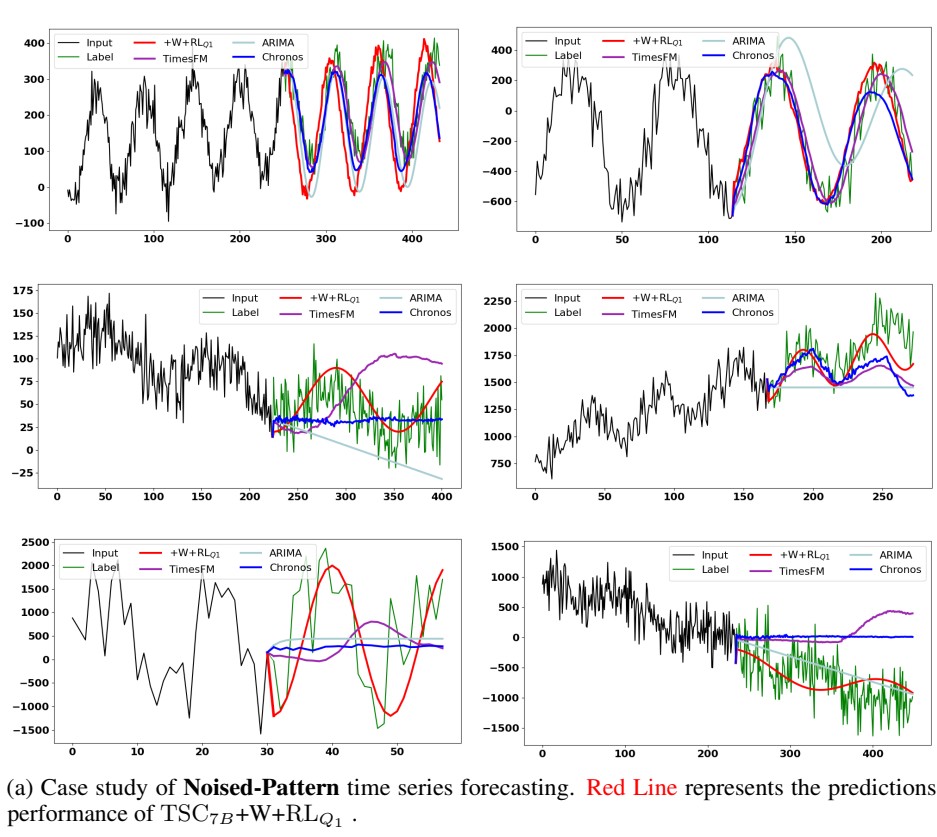

(a) Case study of **Noised-Pattern** time series forecasting. Red Line represents the predictions performance of $\text{TSC}_{7B}+\text{W}+\text{RL}_{Q_1}$ .

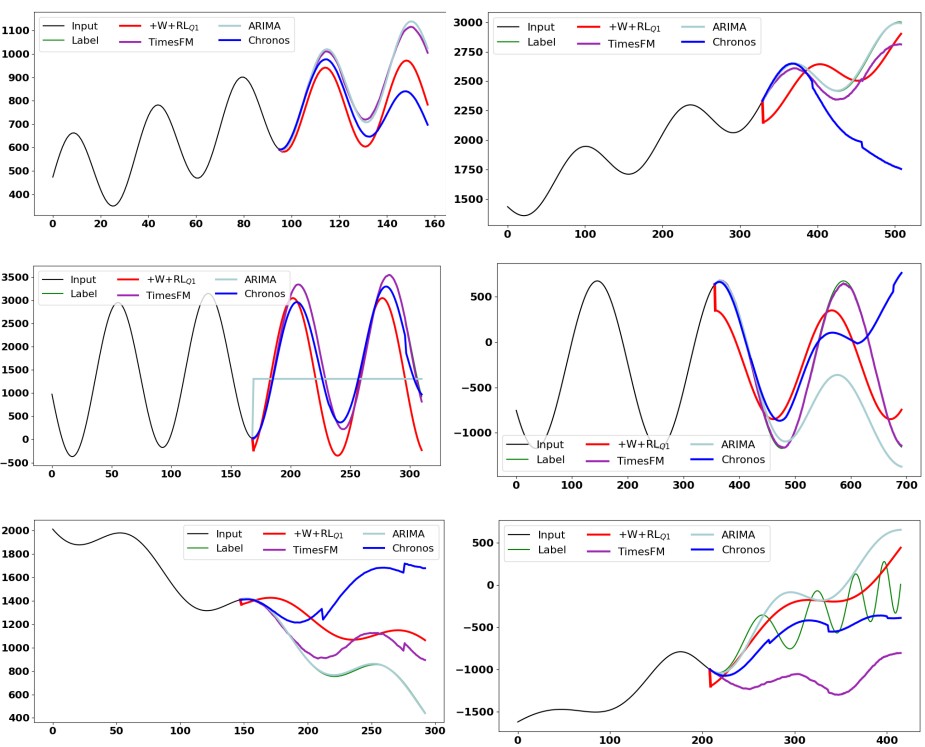

(b) Case study of **Clean-Pattern** time series forecasting. Red Line represents the predictions performance of $\text{TSC}_{7B}+\text{W}+\text{RL}_{Q_1}$ .

Figure 18: (a) is the **Noised-** and (b) is the **Clean-Pattern** cases of time series forecasting on $\text{TSC}_{7B}+\text{W}+\text{RL}_{Q_1}$ , TimesFM, Chronos (Large), and ARIMA.

