# OpenReview forum: "TS2Code: Enhancing Time Series Understanding via Learning to Code"
_ICLR.cc/2026/Conference — ICLR 2026 Conference Withdrawn Submission_

### Official Review · Reviewer_mfMN · 2025-10-20

**Soundness:** 3
**Presentation:** 2
**Contribution:** 3
**Rating:** 4
**Confidence:** 3

**Summary:**

TS2Code attempts to tackle time series reasoning and forecasting by having vision-language models generate code and natural language descriptions from time series images. The idea is creative and the two-stage training (knowledge distillation followed by reinforcement learning) is well-structured. After a careful read, I find some limitations that weaken the impact and practical relevance of this work, but in general, this work is inspiring.

**Strengths:**

a. Interesting Perspective: Using code as an interpretable output for time series tasks is a novel idea and could, in theory, make model reasoning more transparent.
b. Well-Organized Methodology: The two-phase training pipeline is logical and addresses some common issues with shortcut learning.
c. Comprehensive Ablation Studies: The authors provide detailed analyses of code style and training stability, which are valuable from a methodological standpoint.

**Weaknesses:**

a.. Heavy Reliance on Synthetic Data: The overwhelming majority of experiments are conducted on synthetic datasets. While these are useful for controlled analysis, they do not reflect the complexity, noise, and unpredictability of real-world time series. The few real-world datasets included are not sufficiently diverse or challenging.
b. Shortcut Learning Not Fully Addressed: Despite the use of code style constraints, the model appears to still exploit interpolation and value copying strategies. This undermines the claim that TS2Code leads to genuine pattern understanding.
c. Practical Robustness Issues: The paper glosses over how code execution errors, formatting issues, or inconsistencies between code and natural language would be handled in real-world applications. This is a serious concern for deployment.
d. Domain Adaptation and Generalization: There is little evidence that the approach can generalize to new domains, handle multivariate or irregular time series, or work well in low-data regimes.

**Questions:**

1. Can the authors provide a direct comparison with these two recent work, https://arxiv.org/abs/2505.15354   /  https://arxiv.org/pdf/2506.13705
2. Can the authors evaluate TS2Code on more complex, multivariate, and irregular real-world time series, particularly from domains such as finance, healthcare, or industrial IoT—especially in finance or other event-driven datasets?
3. How would the system handle code generation failures, invalid outputs, or inconsistencies in a real deployment scenario?
4. Can TS2Code adapt to new domains or operate effectively in few-shot scenarios? Is the code-based representation actually beneficial for transfer learning?

Note: I would be willing to raise my score if the authors conduct more extensive experiments. It is understandable if the performance does not fully meet expectations, but the paper should present more interesting or insightful conclusions.

---

### Official Review · Reviewer_wrxK · 2025-10-25

**Soundness:** 2
**Presentation:** 3
**Contribution:** 2
**Rating:** 4
**Confidence:** 5

**Summary:**

The paper introduces TS2Code, which feeds time-series images to a VLM to produce both natural-language descriptions and executable Python code; the code, when executed, reconstructs the series, and the reconstruction error provides a verifiable reward for GRPO-based RL that improves the representation space for time-series understanding.

**Strengths:**

The paper presents a well-structured framework that bridges large vision-language models with executable code generation for time-series understanding. By introducing verifiable rewards through code execution and reinforcement learning, it effectively combines interpretability and quantitative supervision. The framework shows consistent improvements across multiple tasks, scales well from 3B to 7B models, and provides an elegant unification of reasoning, reconstruction, and forecasting.

**Weaknesses:**

1.  Most evaluations rely heavily on synthetic or semi-synthetic datasets, which may not capture the irregular sampling, missing values, and high-dimensional correlations present in real-world industrial or medical time series. Demonstrating the model’s robustness on real multivariate datasets (e.g., Electricity, Traffic, Healthcare) or long-horizon forecasting tasks would substantiate the generalization claims.

2.  The teacher model is exposed to ``future images’’ when generating predictive code during the distillation stage. While the paper frames this as a “predictive supervision” setup, the boundary between valid future conditioning and information leakage remains unclear. It would be useful to specify whether the student ever indirectly receives target-domain or future information, and to provide ablations where the teacher’s access to future data is restricted.

3. The main tables report single-run metrics without variance or statistical significance. Including mean $\pm$ standard deviation across multiple random seeds (e.g., five runs) and significance tests (paired $t$-test or bootstrap) would make the empirical claims more reliable. In anomaly detection, the relatively high number of ``invalid answers’’ before warm-up also indicates instability that should be quantified.

4. Large-scale automatic code execution raises practical and ethical concerns, including sandboxing, resource isolation, timeout limits, and blacklisted operations. The paper only briefly mentions success/failure ratios but does not detail safety protocols. Providing these details would enhance the transparency and reproducibility of the pipeline.

**Questions:**

1. Could the authors precisely describe how future images are used during teacher distillation? Are these images completely excluded from the student’s inputs and gradient flow? If future information is partially visible, how does this affect the model’s fairness and generalization in non-stationary domains?

2.  The GRPO framework integrates multiple rewards (Q1 for structure, Q2/Q3 for numeric precision). Have the authors conducted sensitivity analyses or ablations varying their relative weights? Are there trade-offs or threshold effects where emphasizing structural accuracy degrades numeric fidelity or vice versa?

3. In anomaly detection and reasoning tasks, invalid or unexecutable outputs remain frequent. Could the authors analyze whether these arise from syntax errors, numerical overflows, or logic inconsistencies in generated code? Have they explored automatic correction mechanisms (e.g., program repair or constrained decoding) to mitigate such failures?

4. The model relies on image-based time-series representations instead of textual or symbolic forms. Under equivalent capacity and instruction prompts, how does image-based encoding compare with textual embeddings in downstream performance? Clarifying this would highlight whether the visual modality truly contributes additional structure awareness.

5.  Please elaborate on the sandbox environment used for executing generated code. What timeout, memory, and API restriction policies are in place? Are there observed instances of harmful or infinite-loop code generation, and how are they handled in practice?

6. Beyond benchmark datasets, could the authors discuss potential industrial or scientific use cases (e.g., sensor diagnosis, financial forecasting, or medical signal analysis)? Providing even small-scale real examples would demonstrate the practical utility and trustworthiness of executable-code supervision.

---

### Official Review · Reviewer_dytz · 2025-10-28

**Soundness:** 2
**Presentation:** 3
**Contribution:** 2
**Rating:** 2
**Confidence:** 4

**Summary:**

In this paper, the authors proposed an interesting idea, which teaches llm to learn time series patterns (trend and periodicity) through generating code to reconstruct the time series. Firstly, the authors distill knowledge from OpenAI o4-mini to warm up the base model. Then, reinforcement learning is used to further improve the model with GRPO, where reconstruction accuracy and code quality metrics are used as the rewards.

**Strengths:**

S1: The idea of enhancing llm's ability to understand time series through generating code to reconstruct the time series is new.

**Weaknesses:**

W1: Although the idea is new, applying such an idea for time series forecasting and anomaly detection is counterintuitive. I guess humans do not make forecasting and anomaly detection by generating code. The corresponding experiments also verify this. It seems that the forecasting accuracy is even worse than ARIMA by comparing Table 2 and Table 4. The anomaly detection accuracy is also quite bad. Therefore, I think it is not appropriate to use TSF and TSA for evaluation.
W2: The experiments for evaluating the ability of LLM to understand and reason for time series are incomplete. The authors are suggested to include timeseriesexam and MTBench into evaluation. Further, the authors should include more LLM baselines for comparison, e.g. Llama-8B, ChatTime, ChatTS.
W3: The authors use subsets of datasets for evaluation and do not mention whether they will share the model. The provided code for training seems to be not mature. All of the this make the reproducibility of the paper questionable.

**Questions:**

Q1: What does "Unstrc" mean in Table 1? What does "Err" mean in Table 2?

---

### Official Review · Reviewer_LrTa · 2025-10-31

**Soundness:** 1
**Presentation:** 1
**Contribution:** 2
**Rating:** 2
**Confidence:** 4

**Summary:**

This paper introduces TS2Code, a method for training vision-language models to better understand time series data. The approach involves teaching the model to convert a time series, presented as an image, into executable Python code. The core idea is that when this generated code is run, it should reconstruct the original time series. The accuracy of this reconstruction provides a verifiable reward signal, which allows the authors to use Reinforcement Learning (RL) to further refine the model.

**Strengths:**

* Novel proposal on how generating code that generates time-series is useful for learning a representation space.

**Weaknesses:**

* One of the biggest assumptions of this work is that you can write python code that reconstructions the original input. There are many details missing on what exactly the code looks like. For example, is the python function the full generative process, maybe modeled as a SARIMA? How can it be possible to write the generative process for real-world time-series rather than simple synthetically parameterized time-series? I found an example of it in Figure 13, buried in the Appendix, but assumptions about how the generation is done is not described. Additionally, there do not seem to be results in the paper that visualize the said reconstruction. Without in depth exploration of what exactly this python generative code looks like, this paper does not contain sufficient insights that can be useful for the larger research community.
* "Assume you cannot see the future time series image;" Is there anything in the code that enforces this "assumption" from the prompt? how do we know that the LLM is not cheating?
* To be frank, the experimental results in 5.3 and onwards are very difficult to parse and do not set a specific story to understand the relevance of the results. For example, why would it make sense to compare against GPT4o in Table 1 and 2 for reconstruction? GPT4o was not designed for reconstruction. These reconstruction results are also confusing because the paper frames the reconstruction as a means to an end, to help with "understanding" but most of the results do not evaluate the actual understanding.
* Table 3 results seem much worse than GPT4o despite being trained specifically to handle time-series data.

**Questions:**

* Typo in "5.3 IMPORVING REPRESENTATIONS THROUGH RECONSTRUCTION"

---

### Note · Authors · 2026-01-23

I have read and agree with the venue's withdrawal policy on behalf of myself and my co-authors.